# Principal Features of Fatigue and Residual Strength of Composite Materials Subjected to Constant Amplitude (CA) Loading

**DOI:** 10.3390/ma12162586

**Published:** 2019-08-14

**Authors:** Alberto D’Amore, Luigi Grassia

**Affiliations:** Department of Engineering, University of Campania “Luigi Vanvitelli”, 81031 Aversa (CE), Italy

**Keywords:** fatigue, residual strength, fiber composited, constant amplitude loading

## Abstract

This paper summarizes the principal features of composites’ responses when subjected to constant amplitude (CA) cyclic loadings. The stochastic nature of the responses; the absence of a detectable fatigue limit; the sudden drop of strength; the general validity of the strength-life equal-rank assumption (SLERA); and, ultimately, the residual strength-life equal-rank assumption (RSLERA) are discussed on the basis of the selected experimental data available in literature. The objective is defining a robust test in order to ascertain the reliability of the phenomenological models. A two-parameter phenomenological model accounting for the maximum cyclic stress, σ_max_, and the stress ratio, R = σ_min_/σ_max_, was used for guidance through the phenomenology of fatigue. It is concluded that the robustness of the models dealing with fatigue can be checked only when the characteristics of the composites’ responses are described simultaneously with fixed parameters.

## 1. Introduction

The failure of traditional isotropic materials is generally dominated by a single crack nucleation and propagation, while in composite materials, it is attributed to the diffuse damage accumulation resulting from the synergistic effects of different damage mechanisms, eventually occurring at different time and length scales. This complexity lies in the anisotropic and heterogeneous nature of composites, the diversity of constituent materials, and the processing technologies, which make the experimental characterization of composites a complicated task.

The general responses of composite structures to mechanical loadings depend on the environmental conditions (temperature and humidity), the loading rate, the frequency, and loading sequence, and simultaneously taking into account all of the above parameters and simulating their effect on material behavior is impossible in practice. Moreover, the diversity of the material configurations coming from different fiber reinforcing systems, matrices, process technologies, sequences of lamination, and so on, makes the development of a general modelling procedure to cover all these variances difficult. Different from metals where the damage metric, namely the residual strength, is associated with the evolution of a crack length, the absence of a common metric to evaluate the level of damage in composites and their relationship with the degradation of strength or stiffness makes the experimental characterization of a composite material subjected to fatigue loadings mandatory. In other words, the target lifetime of structures subjected to cyclic loading should in some way be predictable, and this requires large laboratory efforts to characterize their fatigue performance. However, any structural component shows unique responses, and this drives researchers to reduce the experimental efforts that still remain large and costly to a minimum, even when few parameters are in play [1,2]. Fortunately, at the level of coupons the mechanical responses of composites to constant amplitude (CA) fatigue loadings show common features, despite the wide materials’ diversity. These features include, but are not limited to, the following: (i) the statistical nature of the responses, (ii) the absence of any detectable fatigue limit, and (iii) the validity of the strength/residual strength-life equal rank assumption (SLERA/RSLERA), stating that samples with a higher static strength have both longer life expectancies and higher residual strength prior to the final collapse. These characteristics will be substantiated in the following paragraphs, and still represent a formidable challenge for any modelling approach, despite their unmistakable simplicity. To this end, a series of models have been developed at a micro- and meso-scale in order to correlate the damage state to the “in service” performances during the fatigue life. However, these approaches also appear to be scarcely reliable for their lack of an extensive experimental database for a given material category. From a different viewpoint, several methods were developed to quantify the properties degradation through directly measurable macro properties, making the phenomenological approaches the most popular methods adopted, at least in industrial environments. The phenomenological models are generally based on stiffness or strength degradation during fatigue, and predictions are based on empirical criteria. However, most of the predictions of the phenomenological models at various stress levels and life fractions were largely unreliable, even though some models sometimes show their applicability to specific test cases. Despite the inconvenience, phenomenological models are used in industrial environments [3,4,5,6,7] as a basis for the structural design of specific components, where the fatigue characteristics of the constituent material are obtained at the level of coupons. This does not exclude that, when applicable, the fatigue models based on a deep understanding of the damage mechanisms correlated to the macro properties may drive more reliable design tools. The phenomenological residual strength models are more popular than those based on residual stiffness, despite the fact that the experimental campaign requires larger efforts, as only one residual strength value can be obtained for one specimen, and the strength degradation rule scatters more seriously than the residual stiffness. In fact, the residual strength has its natural criteria as a failure index, as failure occurs when the residual strength reaches the peak stress, so that the residual strength models are mostly adopted in the industrial environment for safety and reliability purposes [8,9,10,11]. Instead, the mechanistic/hierarchical models [12,13,14,15] require the adoption of specific failure criteria, depending on the representative volume elements (RVE) that, depending on the loading severity and laminate stacking sequence, can be defined at different time and length scales within the same material. However, the hierarchical models have shown their weakness over the years and their reliability appears to be ascertained especially by virtue of the fact that the materials’ diversity makes the approaches limited to the test cases, and scarcely applicable to different materials [15]. The two different approaches, namely the mechanist and the phenomenological one, are not contradictory in the light of the present paper. Here, the relevant features of the composites’ responses to constant amplitude (CA) cyclic loadings are reviewed. The objective is defining a conceptual test campaign for the comparative study of both the phenomenological and mechanistic models. In other words, the robustness of different models should be checked on a common basis, and the mechanistic and phenomenological models’ capability should be ascertained towards the simultaneous prediction of the principal composites’ response features, possibly with fixed parameters. It is believed that once ascertained, the models’ robustness against the constant amplitude (CA) responses they can be eventually used to predict the composites’ responses under variable amplitude (VA) loadings, an issue that remains an open chapter of mechanics of composites. Moreover, we strongly support the idea by Chou and Croman [16], stating that fatigue life models should be particular cases for residual strength models, and that the statistical distribution of the static strength should be reflected into the fatigue life distribution, namely the strength-life equal rank assumption (SLERA). In what follows, we limit our discussion to the experimental data, eventually re-elaborated, available in the literature, and frequently taken as reference data to compare different models [17,18,19]. For the sake of clarity and guidance, the experimental data are modeled by a two-parameter phenomenological model [17], accounting for the maximum cyclic stress, σ_max_, and the stress ratio, R = σ_min_/σ_max_.

## 2. Phenomenology of Fatigue

The stochastic nature of the responses; the absence of a detectable fatigue limit; the sudden drop of strength; the general validity of the strength-life equal-rank assumption (SLERA); and, ultimately, the residual strength-life equal-rank assumption (RSLERA), are the relevant unmistakable characteristics shared by several categories of composites, including short-fiber thermoplastic and thermosetting, continuous mat thermoplastics, glass/epoxy and graphite/epoxy laminates, and notched and un-notched laminates. These principal features are discussed in what follows, on the basis of the selected experimental data available in the literature, along with a two-parameter model recently developed and used in this paper for guidance to illustrate the physics behind the observed responses. 

### 2.1. Static Strength

The large scatter of the static strength is debited, but not limited, to the constituent materials, fibers, resins and prepregs, the process techniques to produce the components, the temperature and moisture effects, and the ply-stacking sequence, respectively. Any kind of composites suffer from the presence of a series of defects like voids, fiber wrinkling, resin-rich zones, dry fiber bundles, and so on, which are unavoidable, despite the efforts spent to standardize the process techniques in order to minimize the sources of defects. We can anticipate that the scatter of static strength reflects the scatter of fatigue life, and so the statistical distribution of the static strength represents the finger print of a given component. Moreover, the inherent existence of defects makes composites far from the ideal one necessarily described in, for example, FEM modelling. For instance, even if delaminations occur to within the resin-rich zone between the plies, the modelling approaches underestimate or frequently neglect the existence of the substantial residual stress that accumulates during the processing stages. The scatter of the ultimate performances (strength and fatigue life residual strength) necessarily reflects the heterogeneous nature of the composites. This is the reason that, different from metals, the development of a single crack is observed under fatigue, and the unavoidable stochastic distribution of voids, fiber misalignments, fiber dewetting, excess resin zones, and residual stresses in the fiber composites determine the development of the diffuse damages with a different origin and location, that randomly accumulate to reach a state of stress, and eventually drive failure. The large variability of strength is reflected in the fatigue response of the structures subjected to spectrum loadings in service. Thus, representing the initial static strength of the composites accounting for the stochastic nature of the events precipitating failure requires the use of statistical distribution functions. Looking at the literature data, the two-parameter Weibull distribution function is largely used to represent the static strength distribution and for scattering studies of any category of composites, even if, especially in the past, both normal and log-normal distribution functions have also been adopted for the same purposes [20]. The Weibull distribution function will be used in the following for both the static strength and fatigue scattering problems, because of its excellent applicability and accuracy. To illustrate, the cumulative distribution function of the static strength is reported in Figure 1 for notched AS4 carbon/epoxy 3k/E7K8 Plain Weave Fabric (open hole specimens) with a 10/80/10 lay-up [21], and un-notched T700S/2500 carbon/epoxy laminate with a (45/0/-45/90)_s_ lay-up [6,22]. The Weibull cumulative distribution function is expressed by the following equation:(1)$$F_{\sigma_{0}}\left(x\right)=P\left(\sigma_{0}\leq x\right)=1-\exp\left[-{\left(x/\gamma \right)}^{\delta }\right]$$
where $F_{\sigma_{0}}\left(x\right)$ is the probability of finding σ_0_ ≤ x, and γ and δ are the scale and shape parameters of the distribution function, respectively. In particular, in Figure 1b, we report both the virtual static strength recovered from the fatigue data, and the experimental static strength. The results of the “calculated” or “equivalent static strength” came from the re-elaboration of fatigue data obtained by a procedure that was illustrated in the literature [9,23,24,25,26,27,28,29,30]. For the sake of completeness, here, we only mention that the robustness of fatigue models is firstly witnessed by their capability to recover the original static strength of the samples subjected to fatigue failure. Among the others, the capability of two such models, referred to as “equivalent strength models”, namely the Sendekyj’s and Caprino’s models [9,23], are compared in Figure 1a, showing substantial equivalence. The original Sendeckyj’s model is used in the following form:(2)$$\sigma_{e}=\;\left[{\left(\frac{\sigma_{r}}{\sigma_{max}}\right)}^{\frac{1}{S}}+\left(n_{f}-1\right)C\right]$$
where *σ_e_* is the strength of a sample, failed under fatigue at a given number of cycles; *n_f_* is shown before starting the test; *σ_a_* is the maximum applied cyclic stress; *σ_r_* is the residual strength; and S and C are the parameters of the Sendeckyj’s model that come from suitably fitting the experimental data. The residual strength models always adopt the same failure criteria, namely, failure occurs when the residual strength degrades down to the value of the maximum applied stress, $\;\sigma_{r}=\sigma_{max}$. With this condition, Equation (2) becomes the following:(3)$$\sigma_{max}\left(1-C+Cn_{f}\right)=\sigma_{u}$$
where *σ_u_* is the static strength. Based on the above equations, a series of equivalent static strength-data points can be obtained, given the experimental S/N data. Then, these data can be ordered and fitted into a two-parameter Weibull distribution, obtaining the life shape parameter as described by Sendeckyj [9]. A similar approach was followed with a different model predicting the fatigue behavior under constant amplitude loadings, and taking into account the loading ratio, R, as follows:(4)$$\sigma_{0N}=\sigma_{0}=\;\sigma_{\max}\left[\alpha \left(1-R\right)(N^{\beta }-1)+1\right]$$

In Equation (4), $\Delta \sigma =(\sigma_{\max}-\sigma_{\min})\;$ is the cyclic stress amplitude, $R=\frac{\sigma_{\min}}{\sigma_{\max}}$ is the loading ratio, N is the number of cycles to failure, α and β are the model parameters, and the original strength of the samples failed under fatigue is $\sigma_{0N}$. Accordingly, the “recovered” static strength distribution, $\sigma_{0N}$, should coincide with the experimentally determined static strength distribution, σ_0_, a case that represents the first requirement for a reliable phenomenological model based on residual strength. 

In Figure 1, the Weibull representation of the “calculated” static strength distribution Figure 1a is reported for the AS4 carbon/epoxy 3k/E7K8 Plain Weave Fabric-open hole specimens, showing the substantial equivalence of Sendekyj’s and Caprino’s models, namely Equations (3) and (4), respectively. In the same Figure 1b, the perfect superposition of the experimental and calculated strength distribution for un-notched T700S/2500 carbon/epoxy laminate with a (45/0/-45/90)_s_ lay-up is reported.

### 2.2. The Strength-Life Equal Rank Assumption (SLERA)

The dispersion of the static strength data should be somehow reflected into the dispersion of the fatigue life data. Indeed, over the years, this nice but vague intuition has been converted to the quantitative assumption that specimens with longer fatigue lives should have had a higher static strength. However, those concepts cannot be proven, as the initial strength of the samples failed under fatigue loading remains unknown. Nonetheless, a quantitative relation between fatigue life and static strength is implicitly included in several models for fatigue [11]. The concept of strength-life correspondence was first introduced by Hahn and Kim [31], and states that the original strength of specimens with a given rank in the fatigue life distribution is equivalent to the strength of the specimens with the same rank in the static strength distribution. Later on, this sentence came to be referred to as the “strength-life equal rank assumption” (SLERA) [16]. Along this line of thought, Barnard et al. [26] argued that the scatter in fatigue life can be shared between a fatigue component and a static one. The scatter due to the static strength is visualized in Figure 2, where real experimental data [6] in terms of static strength and fatigue life under constant amplitude loading (namely, σ_max_ = 290 MPa and loading ratio, R = σ_min_/σ_max_ = 0.1) are reported. In practice, at the extreme tails of the static strength distribution (the realistic extremes of the distribution are taken at 99 and 1 percentile within the distribution and referred to as the upper and lower bounds strength, respectively), namely σ_0UB_ = 530 and σ_0LB_ = 420 MPa, respectively, it results that at σ_max_ = 290 MPa, these samples are in fact tested at different fractions of their strength, σ_max_/σ_0UB_ = 0.55 and σ_max_/σ_0LB_ = 0.69, respectively. Thus, the specimens tested at 55% of their static strength have a longer life expectancy than the weaker specimens, as the severity of loading is lower, despite the fact that the input loading is the same. Given that the severity of the loading affects the life expectancy, the SLERA can be applied only if the scatter properly attributed to the fatigue remains negligible with respect to the static strength distribution effects. The fatigue component of the scatter in fatigue is associated with the growth of the damage that develops during fatigue, and splitting the effect of static strength and the fatigue component within the scatter in fatigue may appear not an easy task, in principle. However, an analytical proof of SLERA was reported by Barnard et al. [26] for different categories of fibre-reinforced epoxy laminates, using a generalized oversimplified model for the S/N fatigue data. In practice, taking into account the 1 and 99 percentile curves as the lower and the upper bound of the static component of the fatigue scatter, respectively, showed that the experimental fatigue data were completely included within the domain confined by the 1 and 99 percentile curves. In the presence of an additional fatigue component, the fatigue scatter bands should result in being greater than those for the static component alone. Thus, the scatter of the fatigue data was attributed to only the variations of the static strength, and no fatigue components were in play. This allowed for an exact correlation between strength and life expectancy.

An important consequence of SLERA is that lowering the applied stress reduces the differences in terms of the severity of loadings between the samples of different strengths, and for this reason, it is expected that the statistical distribution of fatigue life under CA loadings shows a lesser dispersion at a lower maximum stress. To illustrate, we report in Figure 3 the experimental data in terms of fatigue life distribution for T300/934 graphite/epoxy laminates with a (0/45/90-452/90/45/0)_2_ y-up at different maximum stresses, σ_max_ (namely, 400, 345, and 290 MPa), and R = 0. The data are modeled and fully predicted by using fixed parameters of the Caprino’s model [27]. In fact, from Equations (1) and (4), the cumulative distribution function of the number of cycles to failure under given loading conditions was derived as follows:(5)$$F_{N}\left(n\right)=P\left(N\leq n\right)=1-\exp\{-\;{\left[\frac{\sigma_{\max}\;\left[1+\;\alpha (n^{\beta }-1)\left(1-R\right)\right]}{\gamma }\right]}^{\delta }\}$$
where $F_{N}\left(n\right)$ represents the probability to find an N ≤ n. The model’s parameters, α and β, derive from best fitting the fatigue life data and the scale and shape parameters, γ and δ, respectively, of the Weibull static strength distribution, which are experimentally determined or recovered from the fatigue data using Equation (4). It is worth noting that Equation (5) degenerates to Equation (1), namely, the Weibull cumulative distribution function, when n = 1. The empty symbols of Figure 3 represent the experimental data. The filled symbols are the predictions from Caprino’s model and the continuous curves the Weibull representation of the predicted data. To highlight this important feature, the parameters of the Weibull distribution function of the fatigue life data represented in Figure 3 are reported in Table 1. Even if readily apparent in this figure, the numerical values of the shape parameters increase by decreasing the loading severity, a quantitative proof of the lesser dispersion of the fatigue life data at lower maximum stresses, σ_max_.

### 2.3. Fatigue Limit

Different from several metallic alloys, no fatigue limit has been proven to exist in composite materials, despite the superior fatigue behavior of composites with respect to other materials. Indeed, looking at the range of cycles visited during the service life, composite materials show that every load cycle has a potential damaging effect on a composite structure, and should be taken into account in life prediction calculations.

### 2.4. Residual Strength

The residual strength is defined as the strength of the samples fatigued *n* cycles prior failure occurring under a given loading condition. It is observed that the residual strength, σ_in_, degrades smoothly in the first cycles decades [30] from its static value, σ_i__0_, and suddenly drops within a narrow cycle interval until σ_in_ ≈ σ_max_, when the final collapse occurs. This complex behavior can be recovered by observing the experimental data in Figure 4, where the residual strength, fatigue life and static strength data are reported. Fifteen residual strength data were obtained from samples subjected to fatigue for up to 364.000 cycles at σ_max_ = 290 MPa and R = 0, and twenty fatigue life data were obtained under the same condition. First, the dispersion of the residual strength data reflects their statistical nature. Moreover, it can be observed that the “height” of the residual strength distribution is comparable with the “height” of the static strength distribution. Then, the curve describing the residual strength of the stronger sample cannot intersect the residual strength data, as it will be falling on the extreme tail of the distribution of the number of cycles to failure, otherwise it should be admitted to the occurrence of samples with a residual strength higher that their initial static strength, a case with un-physical prerogative (at least for the un-notched samples). Instead, the degradation curve of the stronger sample should overpass such data, and finally suddenly reach the corresponding (highest) number of cycles to failure. The upper dashed line in Figure 4 helps with capturing the described behavior, namely, at a fixed number of cycles, prior to failure, the stronger sample exhibits the highest residual strength. A similar trend can be argued for the weakest sample that falls down to the shortest fatigue life after exhibiting a similar smooth degradation of strength in the first cycles decades. This observation can enlarge the concept of SLERA, namely, the stronger sample should not only have a longer life expectancy, but also a higher residual strength under given loading conditions, and this allows for the modification of SLERA into RSLERA (residual strength-life equal-rank assumption). The above description justifies the concept of sudden death described by Chou and Croman [16,32]. However, this rather complex behavior has rarely bene modeled comprehensively. In their review on the subject, Passipoularidis and Philippidis [11] tested several residual strength models, highlighting that most of the models’ predictions do not conform adequately to the experimental data, showing sometimes dangerous un-conservative predictions. Indeed, the inadequacy of Caprino’s model used so far in evaluating the residual tensile strength of the material after an assigned number of cycles was ascertained long ago [17,18,30]. Therefore, the rate equation on which the model relies requires strong modifications, as reported in what follows, for guidance. We first underline that among the drawbacks of the residual strength models, their deterministic nature prevents predictions in the framework of statistics, that is, the substantial prerogative of the fatigue behavior of composites. Treating the residual strength as a deterministic quantity oversimplifies the approaches and makes the residual strength modeling ineffective when applied to structural design, where, instead, the application of procedures that predicts this statistical behavior is necessary. Moreover, it should be mentioned that simple phenomenological models, namely those with a limited number of parameters, are preferred, because their implementation does not require large experimental data sets. Based on these arguments, an extension of the original Caprino’s model [27] was recently developed [18] in order to fully capture the complex evolution of residual strength. The model is inherently based on the assumption that any specimen has the same rank on the probability distributions of fatigue life, residual strength, and static strength. These assumptions, according to Equation (5) detailed above, can be described as follows:(6)$$P_{REL,\sigma_{max}}\left(X\geq \sigma_{n}\right)=1-P\left(N\leq n\right)=exp\{-\;{\left[\frac{\sigma_{max}\;\left[1+\;\alpha (n^{\beta }-1)\left(1-R\right)\right]}{\gamma }\right]}^{\delta }\}$$
where $P_{REL,\sigma_{max}}$ represents the probability, after a number of cycles *n*, to have strengths $X\geq \sigma_{n}$ under given loading conditions, namely $\sigma_{max}$ and R.

Equation (6) degenerates into Equation (7), as follows:(7)$$\frac{\sigma_{i_{n}}-\sigma_{max}}{\sigma_{i_{0N}}-\sigma_{max}}=exp\{-\;{\left[\frac{\sigma_{max}\;\left[1+\;\alpha (n^{\beta }-1)\left(1-R\right)\right]}{\gamma_{i}\left(\sigma_{i_{0N}}\right)}\right]}^{\delta }\}$$

Equation (7) is obtained by recalling that the residual strength, $\sigma_{i_{n}}$, varies between the static strength, $\sigma_{i_{0N}}$, at *n* = 1, and the maximum applied stress, $\sigma_{max}$, when the number of cycles, *n*, is very close to the number of cycles to failure, *N_f_*, namely *n* ≈ *N_f_*. 

Despite the apparent deterministic nature of Equation (7), the statistical prerogative of the model is recovered by means of Equation (8), as follows:(8)$$\gamma_{i}\left(\sigma_{i_{0N}}\right)=\frac{\sigma_{i_{0N}}}{\sigma (\gamma )\;}\gamma$$
where $\gamma_{i}\left(\sigma_{i_{0N}}\right)$ scales the residual strength of the *i*-th sample with a “virgin” strength of $\;\sigma_{i_{0N}}$, in respect to the reference strength σ(γ) routinely taken at the upper tail of the static strength distribution, and γ is the characteristic scale factor of the static strength distribution, already defined. To illustrate, if the we consider the residual strength curves starting from the extreme tails of the static strength distribution, a domain can be defined confining all of the intermediate residual strength data, given the loading conditions, $\sigma_{max}$ and R, and the number of cycles, *n*. An application of such an approach is reported in Figure 5 for a glass/epoxy composite with a chopped strand mat (CSM), with unidirectionally (UD) reinforced and fabric layers in the following sequence [CSM, fabric, and (CSM and UD)_2_]_s_. The experimental data in terms of fatigue life and residual strength data were taken at different maximum stresses, namely 167 and 283 MPa, and at a different number of cycles [11]. It can be readily seen that the residual strength data remain well confined within the domain defined by the upper and lower bound strength degradation curves, confirming the physical expectations. It is worth mentioning that Equation (7) is fully predictive as the residual strength distribution after *n* cycles, and the fixed loading conditions, σ_max_ and R, can be obtained with no adjustments, with the parameters *α*, *β*, *γ*, and *δ* already being fixed.

### 2.5. The Hierarchy of Damage

The damage in composite materials subjected to cyclic loadings is caused by multiple interacting and/or sequential damage mechanisms, including matrix cracking and crazing, fiber–matrix interface failure, fiber fracture, fiber buckling, and delamination. The schematic of damage development and the corresponding strength evolution under CA loadings is depicted in Figure 6. Three substantial stages of damage accumulation are recognized to be associated with the simultaneous nucleation and growth of different damage mechanisms that may occur at different scale lengths, from the molecular (crack and crazing to within the resin) to the structural scale. As mentioned above, the qualitative damage accumulation described in Figure 6 is a common feature, despite the diversity of composite materials that can be eventually distinguished on the basis of the different extent of the three stages of damage accumulation. From the same figure, it is observed that the correlation between damage and strength is not an easy task, as the strength remains grossly unchanged, even in presence of substantial damage. Nonetheless, the approaches reported in the literature schematically follow the concept of a representative volume element (RVE), which consists of isolating a single degradation mechanism acting at a given length scale, and requiring the definition of sequential hierarchical RVEs to account for the different damage mechanisms developing at different scale lengths. However, despite these efforts, the mechanics of fatigue remain unknown in the framework of mechanistic models, because isolating a single damage mechanism requires oversimplifications that render the correlation with macro properties used in design in vain, for example, strength and stiffness. However, the approaches reported in the literature schematically follow the concept of representative volume element (RVE), which consists of isolating a single degradation mechanism acting on a given scale length, requiring the definition of sequential hierarchical RVEs to take into account the different damage mechanisms developing on a scale of different lengths. However, despite these efforts, the mechanics of fatigue remain unknown in the framework of mechanistic models, because isolating a single mechanism of damage at a time requires excessive simplifications that render the correlation with the macro properties used in the design, such as resistance and stiffness, in vain [1,2,11,12,13,14]. On the other side, from a phenomenological point of view [30,33], Stage I always extends quite along the first one–three cycle decades, depending on the composite category, the damage mechanisms being debited to the nucleation and growth of cracks and crazing to within the matrix rich domains. In these cycle intervals, the damage reaches a critical state that progressively saturates the ability of the matrix to dissipate the mechanical energy associated with the loading severity. Within Stage I, the strength variations are practically undetectable, however. Instead, a measurable strength degradation is observed during Stage II, which extends along several cycle decades. During this stage, strength degradation arises from fiber–matrix interface failure, fiber fracture, fiber buckling, and the incipient delamination. Because the synergistic effects of the different damage that may give rise to ply rupture and delamination at Stage III, strength degradation decreases suddenly until it reaches the maximum applied stress, when the final collapse occurs. However, the damage development described so far, where the damage modes develop sequentially at increasing scale lengths, is rather schematic. The real behavior can be more complex, with the timescales of the different damage modes eventually superposed as described in a recent paper of ours [30]. This was shown analytically in the light of the rate equation derived from the “residual strength model” of Equation (7). The rate equation, given the loading conditions, namely $\sigma_{max}$ and R, and the static strength of arbitrary rank, $\sigma_{i_{0N}}$, was derived [18] as the product of a positive constant, K, and three normalized functions, *f*_1_, *f*_2_, and *f*_3_, as follows:(9)$$\frac{d\sigma_{i_{n}}}{dn}=-K(f_{1}\times f_{2}\times f_{3})$$
where the constuthor Contributionant K is as follows:
(10)$$K=\left(\sigma_{i_{0N}}-\sigma_{max}\right)\left(1-R\right)\alpha \beta \delta \sigma_{max}$$

The functions *f*_1_, *f*_2_, and *f*_3_ are expressed as follows:(11)$$f_{1}\left(n\right)=n^{\beta -1}=\frac{1}{n^{1-\beta }}$$
(12)$$f_{2}\left(n\right)=-\;{\left\{\frac{\left[1+\left(n^{\beta }-1\right)\left(1-R\right)\alpha \right]\sigma_{\max}}{\gamma }\right\}}^{\delta -1}$$
(13)$$f_{3}\left(n\right)=\exp\left\{-\;{\left(\frac{\left[1+\left(n^{\beta }-1\right)\left(1-R\right)\alpha \right]\sigma_{\max}\;}{\gamma }\right)}^{\delta }\right\}$$ and the parameters *α*, *β*, *δ*, and *γ* are already defined and fixed.

The functions *f*_1_, *f*_2_, and *f*_3_ are reported in Figure 7, together with the strength degradation curve, described by Equations (7) and (8), for a T700S/2500 carbon/epoxy laminate with a (45/0/-45/90)_s_ lay-up. We recall that the model described by Equations (7) and (8) is fully predictive, as it requires knowledge of the two model’s parameters obtained from the fatigue life data, namely *α* and *β*, and the parameters of the Weibull distribution of the experimentally determined static strength distribution. The predictions of Figure 7 have been obtained for a sample with an initial static strength of $\sigma_{i_{0N}}=470\;\mathrm{MPa}$, subjected to fatigue at $\sigma_{\max}=200\;\mathrm{MPa}$ and R = 0, with the model’s parameters of *α* = 0.052 and *β* = 0.173, and the parameters of the cumulative distribution function being *γ* = 479 MPa and *δ* = 20.

Three stage of functions’ decay are detectable in Figure 7, with the three functions exhibiting distinct timescales. This conforms to the three-stage hierarchical nature of damage accumulation in the composites, from diffuse matrix cracking (I), to fiber/matrix interface failure (II), to fiber and ply rupture and delamination (III). Thus, despite its phenomenological prerogative, the model adopted here for guidance is able to describe the progression of damage that develops sequentially at three different timescales. The occurrence of timescales superposition, namely the co-existence of different damage accumulation stages debited to the increased loading severity, was fully discussed elsewhere [30], and this makes the mechanistic/hierarchical approaches even more complex, if not impossible.

## 3. Conclusions

The results reported so far outline the principal characteristics of composites’ responses when subjected to constant amplitude loadings. The stochastic nature of the responses; the absence of a detectable fatigue limit; the sudden drop of strength; the general validity of the strength-life equal-rank assumption (SLERA); and, ultimately, the residual strength-life equal-rank assumption (RSLERA) represent a formidable test for the comparative study of phenomenological models. The data were also modeled by a two-parameter phenomenological model, accounting for the maximum cyclic stress, σ_max_, and the stress ratio, R = σ_min_/σ_max_. It was shown that the model captures the substantial features of the composite’s response, including particular insights into the hierarchical damage development. It is argued that only when a model’s robustness is checked against the principal features of the composites’ responses, can they be considered as a candidate to approach the problem of fatigue under variable amplitude loading, an unsolved issue to the knowledge of the present authors. 

## Figures and Tables

**Figure 1 materials-12-02586-f001:**
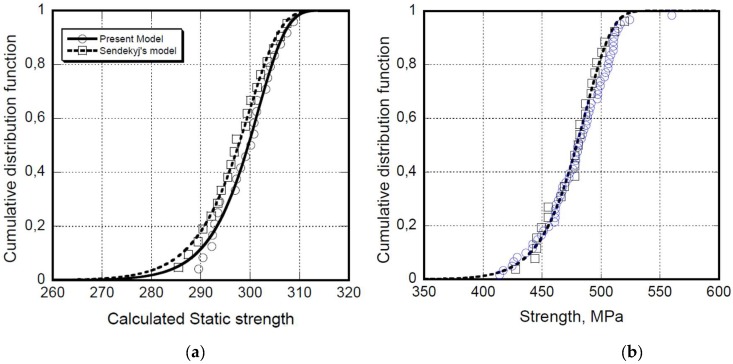
The cumulative distribution function for notched AS4 carbon/epoxy 3k/E7K8 Plain Weave Fabric-open hole specimens (**a**) and un-notched T700S/2500 carbon/epoxy laminate with a (45/0/-45/90)_s_ lay-up (**b**). The Sendekyj’s and Caprino’s models’ equivalence in recovering the static strength from the fatigue life data is shown on the left. On the right, the square symbols represent the experimental static strength data, while the circles represent the nominal strength recovered from the fatigue data using the Caprino’s model.

**Figure 2 materials-12-02586-f002:**
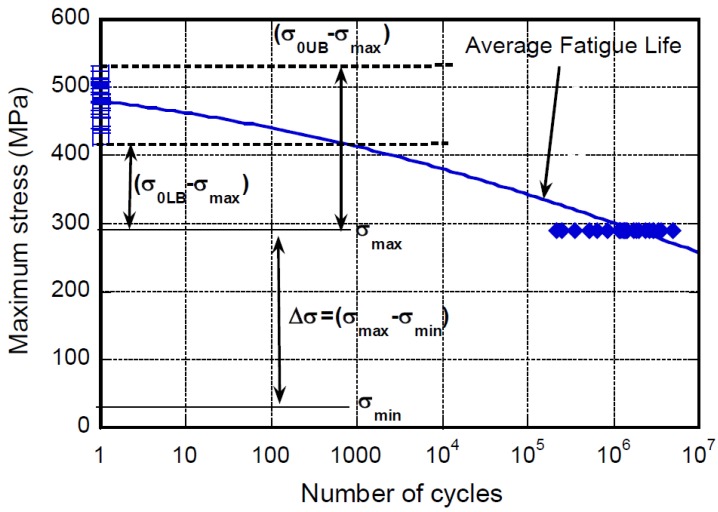
Schematic of the origin of the strength-life equal-rank assumption (SLERA).

**Figure 3 materials-12-02586-f003:**
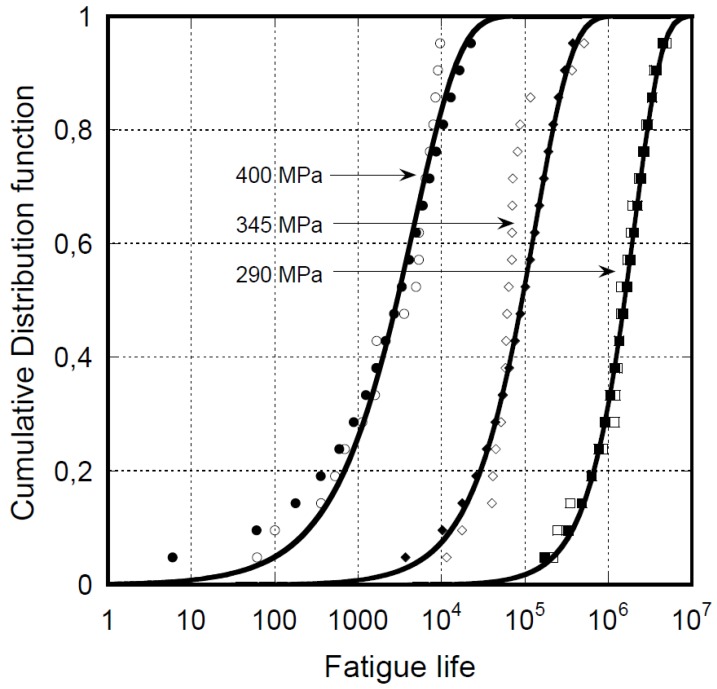
Experimental and calculated fatigue life data for T300/934 graphite/epoxy laminates with a (0/45/90-452/90/45/0)_2_ lay-up, at different maximum stresses, σ_max_, namely 400, 345, and 290 MPa (circle, diamond, and square symbols, respectively), and R = 0. Empty symbols represent the experimental data. The filled symbols are the predictions from Caprino’s model and the continuous curves are the Weibull representation of the predicted data. The experimental data are re-elaborated from the literature [6].

**Figure 4 materials-12-02586-f004:**
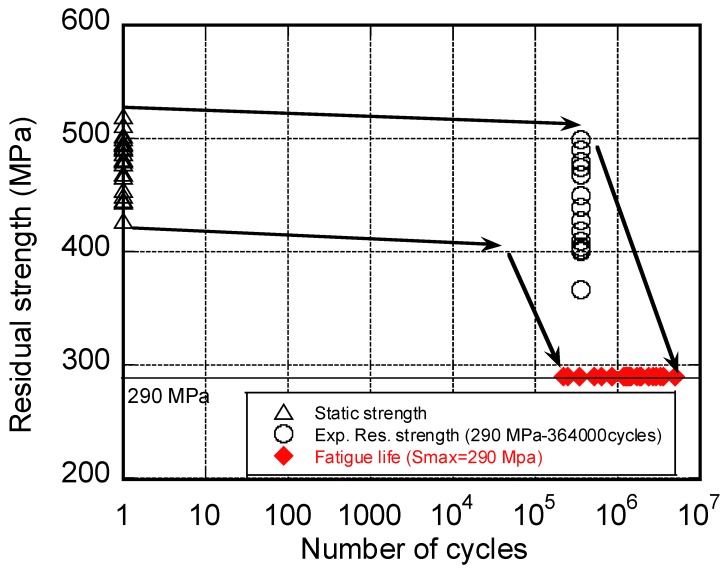
Phenomenological upper and lower bounds of the residual strength evolution. The arrows are guides for the eyes. Experimental residual strengths were measured after 364.000 cycles at σ_max_ = 290 MPa, and stress ratio R = 0. Experimental data re-elaborated from Ref. [6].

**Figure 5 materials-12-02586-f005:**
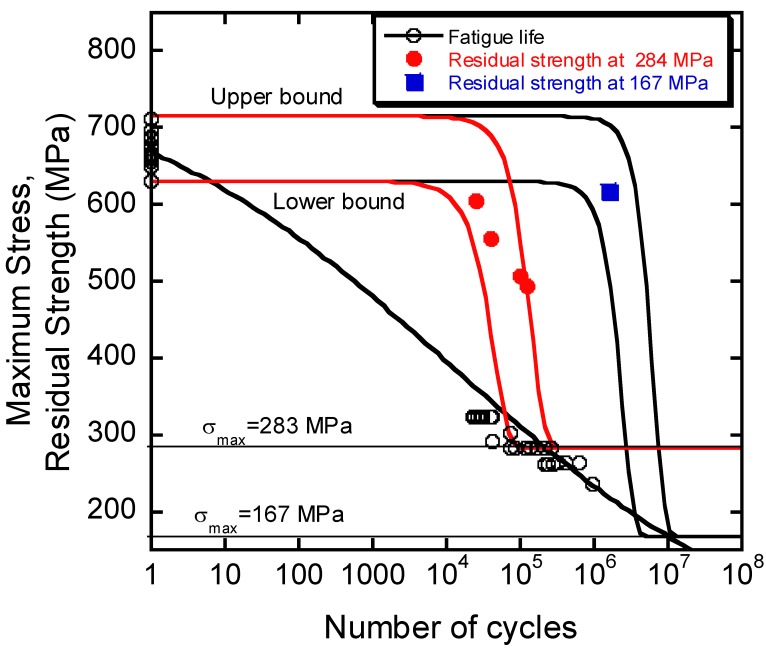
Theoretical upper and lower bounds of residual strength evolution for a glass-fiber-reinforced polyester used as a blade material for wind turbines. The stacking sequence comprised a chopped strand mat (CSM), with unidirectionally (UD) reinforced and fabric layers in the following sequence [CSM, fabric, and (CSM and UD)_2_]_s_. The average volume fraction of fibers was 41%. The experimental data in terms of fatigue life and residual strength data taken at different maximum stresses, namely 167 and 283 MPa, and at a different number of cycles are re-elaborated from Ref. [11].

**Figure 6 materials-12-02586-f006:**
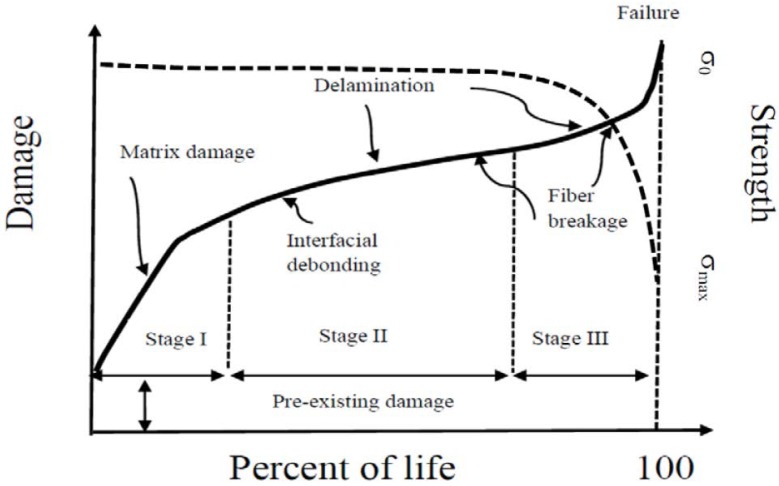
The schematic of damage development and the corresponding strength evolution under constant amplitude (CA) loadings.

**Figure 7 materials-12-02586-f007:**
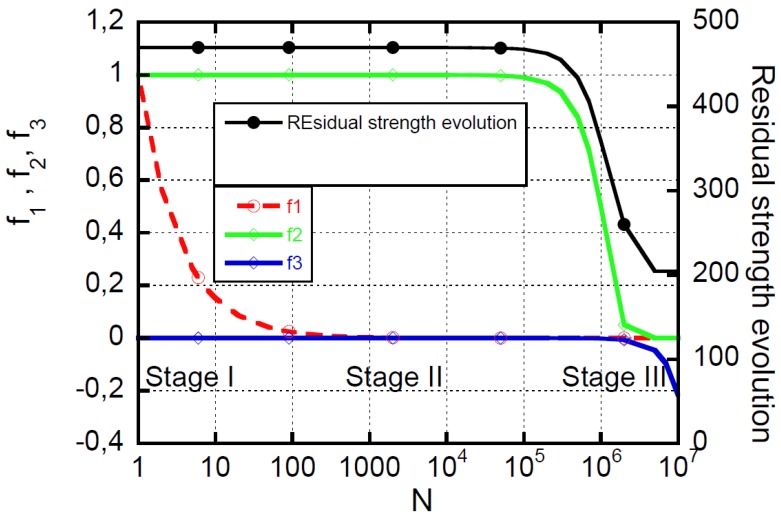
Model predictions of the three decaying functions (*f*_1_, *f*_2_, and *f*_3_) and the residual strength evolution for T700S/2500 carbon/epoxy laminate with (45/0/-45/90)_s_ lay-up.

**Table 1 materials-12-02586-t001:** The Weibull parameters for fatigue life.

Max. Stress MPa	Shape Factor	Scale Factor
290	1.337	2.06 × 10^6^
345	0.987	1.33 × 10^5^
400	0.78	4.68 × 10^3^

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
