# Peer review of "Principal Features of Fatigue and Residual Strength of Composite Materials Subjected to Constant Amplitude (CA) Loading"

_materials, 2019, doi:10.3390/ma12162586_

Round 1

Reviewer 1 Report

The paper titled as “Principal features of fatigue and residual strength of composite materials subjected to constant amplitude (CA) loading” appears as very interesting and deserve being published in Materials.

In essence, the authors have used data in literature to study the phenomenology of fatigue in terms of the static strength by using two  “equivalent strength models” for published data of epoxy/carbon fiber laminates both notched and unnotched.

The dispersion of these data are considered by the using of SLERA assumptions by using published data for another type of graphite/epoxy laminates, and these data are fully modeled by using parameters of the Caprino´s model. The prediction appears to be very good.

By using data of  a type of glass fiber/epoxy composite, and taking in mind the limitations of the Caprino´s model, a phenomenological upper and lower bounds of the evolution of the residual strength is presented jointly with the theoretical bounds for another glass/ polyester as the used in turbine blades. The results obtained are very impressive.

Based on a carbon epoxy laminates, the hierarchy of the damage is described.

All the above viewpoints of the authors about the problem and solutions of the modeling of fatigue and residual strength under CA conditions are of a key importance to be taken in mind when designing with composite materials (In the case of the authors, when using thermoset based composites).

Finally, just a suggestion, although the authors include in the text that the thermoplastic based composites also present stochastic nature of the responses, the absence of detectable fatigue limit, the drop of strength and the general validity of the SLERA and RSLERA approaches (likely the thermoset based composites), they have only used data of themoset based composites in the article. So, as a suggestion, this reviewer would like to mention that the use of the word “composite materials” without specifying the type studied may be corrected by replacing  “composite materials” by “thermoset composite materials”, in order to avoid confusion to any potential reader of MATERIALs, not necessarily expert in organic based composite materials.

This reviewer has enjoyed very much of the reading of the article, the accurate identification of the problems, the conclusions, et cetera. So, the paper deserves being published since it offer a high scientific standard, and probably would become a reference in the field due to the clearness of the concepts as described by the authors. In essence, the article is very useful not only from its scientific quality, but also from the ability of being a tool for educative purposes.

So, consequently, this reviewer strongly recommends the publication of the article with the only amending in the title as proposed above these lines.

Author Response

We agree with the reviewer's comments. Accordingly the manuscript was amended with the small corrections required.

Reviewer 2 Report

This manuscript is well written and authors described in detail the ongoing problem to accurately predict the fatigue behavior of the composite with confidence. The have represented the two parameter phenomenological model that will account the fatigue behavior of the composites and found the robust model. I personally liked this manuscript as we face the problem of determining the fatigue behavior in my lab. I believe this manuscript will help the other researchers. However, I just recommend minor addition of detail described below.

1.     In the introduction, I will recommend elaborating more on the residual strength and stress as it set the foundation for the phenomenological and mechanistic model. [line 65-74]

2.     Add some more detail about inadequacy of Caprino’s model for evaluating residual strength of material [line 263]

Author Response

I agree with the reviewer's comments. Accordingly I modified the manuscript where he/she suggested.